# Illness narratives and chronic patients' sustainable employability: The impact of positive work stories

**Inge M. Brokerhof** [1]*, **Jan Fekke Ybema**[2], **P. Matthijs Bal**[3]

1 Department of Management and Organisation, Vrije Universiteit Amsterdam, Amsterdam, The Netherlands,
2 Department of Social, Health and Organisational Psychology Utrecht University, Utrecht, The Netherlands,
3 Lincoln International Business School, University of Lincoln, Lincoln, England, United Kingdom

* i.m.brokerhof@vu.nl

**Data Availability Statement:** Data cannot be shared publicly because of personal information of the participants, who were promised that their data would be treated confidentially. Data set and syntax will be uploaded via Darkstor and will be available

## Abstract

The number of workers with a chronic disease is steadily growing in industrialized countries. To cope with and to give meaning to their illness, patients construct illness narratives, which are widely shared across patient societies, personal networks and the media. This study investigates the influence of these shared illness narratives on patient's working lives, by examining the impact of reading a positive work story versus negative work story on patients' sustainable employability. We expected that this relationship would be mediated by positive emotions and the extent to which the story enhanced awareness of desires future selves, and moderated by identification with story character. An online field experiment with 166 people with Inflammatory Bowel Disease in The Netherlands showed that while reading a positive story of a patient with the same condition significantly increased positive emotions, these emotions did not influence sustainable employability. However, reading a positive story was related to higher sustainable employability when patients became more aware of their desired possible future work selves. Finally, identification with the story character moderated the impact of story type on sustainable employability. This study showed that personal engagement with a positive work story of a fellow patient is related to higher sustainable employability. Findings can be helpful for health professionals to empower employees with a chronic disease.

## Introduction

*I see a positive future for myself! I know I have the brain and perseverance to hold a challenging position that mentally demands a lot from me. I am limited by my body to work part-time, but I find it nice to use that extra day off to do something fun (. . .) I will never stop learning in my career. (. . .) PS. I took a photo of this as a memory for myself for when I am not doing so well:)*

*- Participant, positive story condition*

from the Vrije Universiteit Institutional Data Access via data manager Veerle Eggens (v.r.c.eggens@vu.nl) or Inge Brokerhof (corresponding author) for researchers who meet the criteria for access to confidential data.

**Funding:** The authors received no specific funding for this work.

**Competing interests:** The authors have declared that no competing interests exist.

During the last decades the prevalence of different types of chronic illnesses have rapidly increased [1] causing chronic illness to be labeled as a new global health burden [2]. Chronic illnesses can be defined as "illnesses that are prolonged, do not resolve spontaneously, and are rarely cured completely" [3],[4]. Living with a chronic illness has a pervasive effect on various different aspects of people's personal lives [5].

Besides physical distress and coping with the direct symptoms of the illness, chronic illnesses affect people's identity, future expectations, employability, working life, social life and mental health [6–8]. While employment may pose extra challenges for people with a chronic illness, having work is important as it provides both manifest and latent benefits: work does not only provide income and a means for living, it also provides latent benefits such as a daily time structure, contributing to a collective purpose, social contacts, social status and activity, which in turn contribute to psychological well-being [9]. Reversely, being unemployed is associated with a lack of these latent benefits resulting in distress [10] and lower mental health and well-being [11, 12].

Since the number of workers with a chronic disease is steadily growing in industrialized countries [13], this draws attention to their specific workplace problems and their sustainable employability, which we define as the perceived ability and motivation to maintain a healthy working life until retirement age [14]. Sustainable employability has been conceptualized as a multi-faceted construct [15], yet at the core is a mindset that is focused on long-term employment and the subjective perceptions that one will remain able and motivated to work until retirement age [14, 16]. This perception will increase chances of successful employment as employees with a high sustainable employability are motivated to actively manage their work and balance it with their personal values and capabilities [14, 17]. Due to increased prevalence of chronic illnesses and the additional challenges of working with a chronic illness poses for sustainable career development, research on interventions to increase sustainable employability of chronic patients is relevant for both society at large and these individuals themselves in order to maintain a healthy and productive workforce [18, 19].

One way to make sense of and cope with the impact of chronic illness is through creating an illness narrative, which is "a story the patient tells, and significant others retell, to give coherence to the distinctive events and long-term course of suffering" [20–22]. Illness narratives are often shared and available to large audiences through social media, television, and patient support group websites and magazines. Stories about illness thereby contribute to a broader, cultural understanding and the social construction of the illness [23–25] which in turn influences patients [20, 24] and medical professionals [26].

Even though the construction of personal illness narratives is undergirded by larger cultural narratives and the social construction of the illness [20, 25, 27], little is known about the direct impact of illness narratives on fellow patients. These narratives can provide patients with information and examples of how to cope successfully with chronic illness [28]. Sometimes patients may actively seek information via shared illness narratives of fellow patients as part of the coping strategy of information seeking [6, 29], illustrative of a general trend where patients engage more in medical self-help online than seeking care from medical professionals [28].

The current experimental field study aims to add to the existing body of research on illness narratives by investigating the impact of illness narratives on fellow patients, specifically the effect of reading an illness narrative about working with a chronic condition on patients' sustainable employability. Principally, the impact of positive, hopeful stories, versus negative, depressing stories is investigated. In this context, "positive" does not mean that the patient has not suffered from the illness for a prolonged time or has only mild symptoms, yet it means that the story character is coping well and has found a way to successfully work, despite of her/his illness. "Negative" stories refer to illness narratives in which the patient is depressed and is not

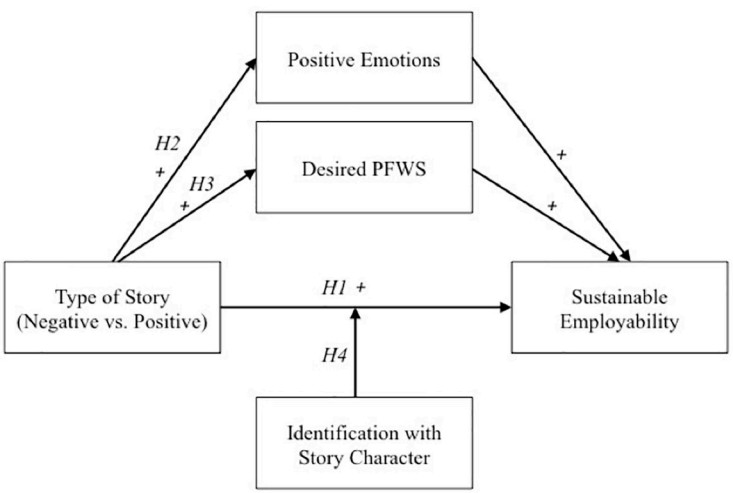

PFWS = Possible Future Work Self.

**Fig 1. Conceptual model of the impact of story on sustainable employability.**

coping well with the illness. Positive illness narratives could offer people inspiration, help them with visualizing a positive work future or provide role models for building a meaningful life. Compared to negative illness narratives, positive stories could therefore stimulate patients to imagine a sustainable career despite their illness.

Besides the direct impact of positive illness narratives on sustainable employability, we propose that positive stories will increase patients' sustainable employability by evoking positive emotions via the mechanism of emotional contagion and subsequent broaden-and-build processes that increase the thought–action repertoire in readers helping them envision more possibilities for their future working life [30, 31]. Secondly, we expect that positive illness narratives will increase the awareness of the patient's own wished-for future, by inspiring their desired Possible Future Work Self (PFWS)–an "individual's representation of himself or herself in the future that reflects his or her hopes and aspirations in relation to work" [32]. Lastly, we expect that through social comparison mechanisms [33] the reader's identification with the story character will moderate the impact of story type on sustainable employability, with stronger relationships when identification with a story character is high rather than low. Fig 1 summarizes the conceptual model investigated in this study.

## Narrative impact

A large body of studies shows that narratives have a profound impact on people [34], influencing people's world-beliefs [35], enhancing social skills, such as empathy [36, 37] and increase self-awareness and self-reflection, ultimately leading to changes in people's sense of self [38, 39]. Recently, the persuasive power of narratives has been applied in medical contexts [40], where narrative methods were used for patient communications [41–43]. These narratives influenced patients' decision-making processes regarding different treatment options. A variety of narrative impact mechanisms in medical contexts has been found, ranging from narratives increasing attention for and comprehension of medical information [44], narratives inducing decision-making bias by making people rely on heuristic rather than systematic mental processes [45], or, contradictorily, narratives have shown to stimulate patients to reflect on

their own impact bias, thereby helping them overcome barriers for cancer screening, for example by making them aware of their own affective forecasting [40].

From a career perspective, personal narratives about work and building a career also provide inspiration and guidance to workers [46]. Narratives of successful business people are regarded as important tools for successful interventions in career counseling [46, 47]. This may be even more relevant for people with a chronic illness, for whom building a successful career poses additional challenges [19]. Indeed, personal illness narratives shared in support group settings helped people cope with their illness in different domains of life, including family and working life [48].

Encountering a positive work narrative may increase hope and work motivation in fellow patients, by showing them the possibility of an open future with new possibilities despite of illness. This perception of an open-ended future could, according to socio-emotional selectivity theory [49] stimulate patients to engage more in planning, future-oriented behavior and goal-setting. Moreover, in the context of the shifting perspectives model of chronic illness [50] providing people with a positive future work perspective can make patients shift from an illness perspective to a wellness perspective, whereby the focus of the patient is not only on managing the illness, but also on achieving other important life goals, like having a valuable working life.

Proactive behavior, motivation and adaptability for maintaining a healthy and satisfying future working life, are important facets of sustainable employability [14]. Workers who perceive themselves as sustainably employable are actively motivated in managing and balancing their work to their personal values and capabilities [14, 51]. For chronic patients it is important to be motivated to actively deal with health issues regarding work, which is a key factor to having a healthy, successful and fulfilling working life [19]. Since positive stories offer an empowering experience, increasing a patient's positive future time perspective, we expect that positive work stories will be positively related to sustainable employability compared to negative work stories of patients with the same chronic illness.

*Hypothesis 1*: *Reading a positive work story of a fellow patient is positively related to sustainable employability of chronic patients compared to a negative story.*

## The mediating role of positive emotions

Emotions play a central role in the construction and experience of stories [52]. Narratives can evoke strong emotions, experienced similarly to emotions in the real world [34]. By means of emotional contagion mechanisms, which pose that people adopt the emotions of their surrounding environment [31], patients reading a positive illness narrative will also likely adopt the emotional sentiment in the story and experience emotions like joy, inspiration and comfort. Consequently, the broaden-and-build theory of positive emotions predicts that this positive emotional mind-set will broaden their "momentary thought–action repertoire" increasing creative thinking and generating new solutions [30]. After reading a positive work story of someone with the same illness, patients are therefore more likely to perceive ways to achieve sustainable employability vis-à-vis patients who are reading negative stories. Therefore, we expect that the experience of positive emotions will mediate the relationship between positive work stories and sustainable employability.

*Hypothesis 2*: *Positive emotions mediate the relationship between reading a positive work story and sustainable employability.*

## The mediating role of possible future work selves

Narratives offer people opportunities to experiment with different possible selves [53, 54] and different lives [55]. Therefore, patients who put themselves in the shoes of a character with a positive work experience may be stimulated to imagine their own desired or wished-for future work selves. Strauss and colleagues [32] found in a series of studies that awareness of Possible Future Work Selves (PFWS) increased proactive career behavior and served as an important motivator for planning a career. Proactivity and awareness of sustaining a career in the future are key concepts of sustainable employability [14]. We therefore expect that awareness of desired PFWS will mediate the relationship between reading a positive story and sustainable employability. Reading positive stories will positively relate to people's sustainable employability if it helps them imagine themselves in a desired future career.

*Hypothesis 3*: *Possible Future Work Selves mediate the relationship between reading a positive work story and sustainable employability.*

## The moderating role of identification with narrative role models

An important mechanism by which stories influence people concerns the reader's identification with the main story character [36, 56]. When identification with the main character is stronger, stories generally have more impact [56]. Based on social comparison processes [57] we expect that identification with the main story character will moderate the relationships between reading a positive story and sustainable employability.

The Identification-Contrast model [33, 58] argues that social comparison with others who are doing better or who are doing worse may either improve or worsen well-being and self-evaluation depending on whether individuals identify or contrast themselves with the other person. If individuals identify with a better-off other this will lead to inspiration and improved well-being due to assimilation of the other's position, whereas contrast with such a successful person would lead to dissatisfaction and lowered self-views [59]. On the other hand, contrast with worse-off others may lead to consolation about one's own more favorable position whereas identification may lead to fear that one would deteriorate to the other's worse position. Based on this reasoning, we expect that a positive story will only lead to more positive self-views (i.e., higher sustainable employability) in comparison to a negative story if readers identify (rather than contrast themselves) with the main character.

*Hypothesis 4*: *Identification with the story character moderates the relationship between positive work story and sustainably employability, such that a positive story only leads to higher sustainable employability when identification is high.*

# Methods and materials

## Participants

An online field experiment was conducted in The Netherlands with a sample ($n$ = 166) of people with Inflammatory Bowel Disease (IBD). IBD is a chronic disease with the two most common diagnoses being Crohn's disease and ulcerative colitis. Even though many people with IBD are able to have some form of employment [60] the uncertainty of the disease, which is characterized by active periods with flare-ups and quiet periods of remission, has a strong psychological impact on people suffering from it [61].

This study was approved by the ethical review board of Utrecht University, the Netherlands. Patients were informed about the experiment in the digital newsletter of the Dutch IBD foundation, CCUVN (Crohn en Colitis Ulcerosa Vereniging Nederland), where they could click on a link to participate. Participation in the study was voluntarily and participants were told that they could stop at any time if they did not wish to continue. Informed consent was obtained from each patient before starting the experiment. At the end of the experiment participants had the opportunity to enter in a prize raffle, in which they could win one of two 50-euro digital shopping vouchers. Initially, 252 individuals clicked on the link provided by CCUVN. In total, 186 of this group finished the experiment. Participants who dropped out of the study did not significantly differ from the final sample. After removing 14 people from the analysis because they failed the manipulation check (a simple question to check whether they had read the story) and 6 people who were retired, the final sample consisted of 166 participants.

## Design and procedure

The online field experiment consisted of two experimental conditions: reading a positive work story of a fellow patient versus reading a negative work story of a fellow patient. Of the 166 participants who completed the experiment, 90 participants (54%) were randomly assigned to the positive story condition and 76 (46%) were put in the negative story group.

First, all participants filled in demographics, questions about work and mental and physical health, mastery, employment status and type of IBD. Subsequently, participants were randomly assigned to either the positive story condition or the negative story condition. After reading this story, participants first filled in the manipulation check, which was a relatively easy question the story content (i.e., "how long ago was the main character diagnosed with IBD?") to make sure they had read the story attentively. Then they answered questions about their identification with the character and their affect after reading. After this, the participants described their Possible Future Work Selves in an open question and rated their own PFWS. After the PFWS measure, participants answered a quantitative scale of sustainable employability. To counter-balance a possible negative impact of the story, a positive mood manipulator was added for all participants at the end of the experiment, where participants were encouraged to write three good qualities of themselves and three people who socially supported them.

## Measurements

Two different stimuli were used in the experiment, a positive illness narrative for the positive story condition and a negative illness narrative for the negative story condition. The narratives were fictitious, but based on real experience since they were written by a short story author with IBD, therefore a fellow patient of the participants. The stories were written in Dutch, they were of equal length and were gender neutral, and were written from a first-person perspective since this is found most effective in an illness context [45]. The narratives were similar in terms of illness onset, diagnosis and history of illness, where in both stories the patient had struggled severely and for a prolonged period of time with health issues, also for several years before the official diagnosis (narratives are available in S1 File).

In the story, the main character was diagnosed with IBD five years ago, and reflects on living with IBD, specifically focusing on working life. Yet the tone and ending of the story is either positive (after five years the main character is coping well, has found a good balanced job) or negative (after five years the main character is not coping well and has tried working but is currently unemployed).

**Positive emotions.** "Participant's emotions after reading the story were measured by a list of 12 positive and 12 negative emotions, whereby people were asked to select all emotions that they currently experienced, based on the Multiple Affect Adjective Checklist (MAACL, [62, 63]. Examples of items referring to positive emotions were "encouraged", "inspired" or "comforted", and examples of items referring to negative emotions were "sad", "discouraged", "irritated" or "anxious". The positive emotions captured two constructs, namely: comfort and inspiration, whereas the negative emotions captured three constructs, namely: depression, hostility and anxiety. In the present study only positive emotions were examined, and comport and inspiration were combined into one score for positive affect."

**Desired possible future work selves.** Participants read a short description of PFWS and were asked to freely describe their PFWS in an open question (question text is available in S2 File). On average, participants spend 276.8 seconds on thinking about and writing their PFWS, with a large variety between people ($SD$ = 297.2). Similar to the method employed by Strauss and colleagues [32], participants were then asked to self-rate their micro-narratives. They rated on a 5-point Likert scale (ranging from 1 = "Absolutely not" to 5 = "Very much") to what extent they had just described a desired self, which was used as an indicator of how aware patients were of their desired PFWS at that moment.

**Identification.** The measure for identification with the main character consisted of three items [63], which were averaged to a total score ($M$ = 3.23, $SD$ = 1.04, $\alpha$ = .87). A sample item is "Can you recognize yourself in the main character?" (Responses on a 5-point Likert scale, ranging from "No, absolutely not" to "Yes, absolutely").

**Sustainable employability.** For sustainable employability ($\alpha$ = .89) a six-item scale was developed, based on earlier research [64] and comparable to the conceptualization of Le Blanc and colleagues [16]. Several pilot studies with this scale suggested good reliability ($\alpha$ > 0.75) and meaningful relationships with other constructs [65]. A sample item is "I expect that, until retirement age, I will be motivated to work" (Responses on a 5-point Likert scale ranging from "Definitely not" to "Definitely"). For a full overview of all items, see S3 File.

**Physical and mental health.** The control variables for physical and mental health were measured with the SF12 [66] and scored into a physical health score ranging from 13.74 to 61.67 ($M$ = 44.8; $SD$ = 10.08) and a mental health score ranging from 22.27 to 60.83 ($M$ = 45; $SD$ = 10). The measure consists of 12 different items and is approximately orthogonally scored, using the original scoring [66], with questions about both physical and mental health. For example: "During the past 4 weeks, how much did pain interfere with your normal work (including both work outside the home and housework)? (Responses on a 5-point Likert Scale ranging from "Not at all" to "Extremely"). The measure of physical health was used as a background variable, to check whether there would be differences between conditions. Mental health was added to the model as a control variable, since it is strongly linked to proactivity and motivation at work [67].

**Mastery.** Mastery was measured with the Pearlin Mastery Scale [68], which consists of seven items, which are answered with a 5-point Likert scale. Sample items are: "I often feel helpless in dealing with the problems of life" (reversely scored) and "What happens to me in the future depends on me". Mastery ranged from 1.86 to 5 ($M$ = 3.45; $SD$ = .66). Since mastery is linked to resilience and motivation [69], mastery was added as a control variable to the model.

## Analysis

For the statistical analysis SPSS 24 was used with the PROCESS plug-in of Hayes [70] to be able to test all hypotheses simultaneously in one model. In order to generate bias-corrected

confidence intervals, correct for irregular shaped sampling distributions and increase the robustness of the indirect effect, the standard bootstrap option of 5,000 resamples in PROCESS was used [71]. In the moderation analysis with Process, identification was mean-centered. The story condition was dummy-coded (0 = negative story; 1 = positive story) in all analyses.

## Results

### Descriptive statistics

To check for significant differences in demographic and background variables per condition an ANOVA was carried out, which showed no significant differences between conditions. The two most common diagnoses in the sample were Crohn's disease (55.4%) and ulcerative colitis (43.4%). 33.7% of the participants were aged between 18 and 34, 37.3% were between 35 and 49, and 28.9% were age 50 and higher. 34.3% were diagnosed less than 5 years ago, 39.8% 5–15 years ago, and 25.9% longer than 15 years ago. 82.5% were either employed or self-employed and the remaining 17.5% were unemployed, disabled for work, or student. See Table 1 for a correlation table of the descriptive statistics.

### Hypotheses testing

Similar to the model, Hypotheses 1–4 were simultaneously tested using PROCESS. Table 2 shows the results of the bootstrapped moderated mediation regression analyses for the direct impact of the story condition, the indirect effect of story condition through positive emotions and desired PFWS on sustainable employability, and the moderation of identification with the main character. We ran the analyses whilst controlling for mental health, and mastery. Age was also added as a control variable, because it is intrinsically linked to sustainable employability [16].

Overall, the final model predicted 34% of the variance in sustainable employability. H1 predicted that reading a positive work story of a fellow patient increases sustainable employability of chronic patients compared to a negative story. However, no main effect for story condition was found ($b$ = -.05, $ns$). H1 is therefore rejected.

**Table 1. Means, standard deviations, reliabilities and correlations of the study variables.**

| Variable | M | SD | 1 | 2 | 3 | 4 | 5 | 6 | 7 | 8 |
|---|---|---|---|---|---|---|---|---|---|---|
| 1. Gender | .27 | 44.60 | - - | | | | | | | |
| 2. Physical health | 44.8 | 10.08 | -0.08 | - - | | | | | | |
| 3. Mental health | 45.0 | 10.00 | -0.16* | 0.21** | - - | | | | | |
| 4. Mastery | 3.45 | 0.66 | -0.14 | 0.45** | 0.54** | .84 | | | | |
| 5. Desired PFWS | 3.83 | 1.03 | -0.19* | 013 | 0.16* | 0.14 | - - | | | |
| 6. Positive Emotions | 2.08 | 2.13 | -0.11 | 0.17* | 0.06 | 0.24** | 0.17* | .68 | | |
| 7. Identification with story character | 3.23 | 1.04 | 0.09 | 0.03 | -0.17* | -0.09 | 0.20* | 0.42** | .87 | |
| 8. Sustainable Employability | 4.01 | 0.71 | -0.11 | 0.31** | 0.33** | 0.43** | 0.29** | 0.20** | 0.15 | .89 |

Reliabilities are reported along the diagonal. N = 166. PFWS = Possible Future Work Self. Gender 0 = female, 1 = male.

† $p < 0.1$

* $p < 0.05$

** $p < 0.01$

*** $p < 0.001$.

**Table 2. Bootstrapped mediation regression analysis predicting sustainable employability.**

| Variable | Positive Emotions | Desired PFWS | Sustainable Employability |
|---|---|---|---|
| *Control Variables* Age | -0.13 (0.19) | -0.31 (0.10)** | -0.15 (0.06)* |
| Mental health | -0.01 (0.02) | 0.02 (0.01)† | 0.01 (0.01) |
| Mastery | 0.99 (0.27)*** | 0.09 (0.14) | 0.27 (0.09)** |
| *Independent Variable* Story type | 1.58 (0.30)*** | 0.45 (0.15)** | -0.05 (0.11) |
| *Pathway 1* Mediation of Positive Emotions | | | -0.01 (0.03) |
| *Pathway 2* Mediation of Desired PFWS | | | 0.11 (0.05) * |
| *Pathway 3* Direct effect of Identification Moderation of Identification with Story Character | | | -0.09 (0.08) 0.42 (0.12)*** |
| Intercept | -1.31 (0.98) | 3.15 (0.50)*** | 2.59 (0.36)*** |
| F R² | 10.24*** 0.20 | 6.13*** 0.13 | 10.32*** 0.34 |

N = 166. PFWS = Possible Future Work Self. For every parameter the unstandardized coefficient (B) is reported with in brackets the corresponding standard error (SE). For bootstrapping 5,000 resamples were requested.

\* $p < 0.05$

\*\* $p < 0.01$

\*\*\* $p < 0.001$.

H2 predicted that positive emotions mediate the effect of reading a positive story on sustainable employability. The results showed that reading a positive story indeed led to significantly more positive emotions than the negative story ($b = 1.58$, $p < 0.001$). Table 2 shows that positive emotions did, however, not significantly predict sustainable employability ($b = -.01$, *ns*). Unstandardized indirect effects of story condition on sustainable employability through positive emotions were computed for each of 5,000 bootstrapped samples, and the 95% confidence interval was computed by determining the indirect effects at the 2.5th and 97.5th percentiles. The bootstrapped unstandardized indirect effect of H2 was -.018 (*ns*; 95% CI = -.10 to .06). Therefore, H2 was rejected.

H3 predicted that a positive story increases sustainable employability through desired Possible Future Work Selves. The results showed that reading a positive story indeed significantly increased participant's awareness of desired PFWS ($b = 0.45$, $p = 0.004$). Subsequently, increased awareness of desired PFWS also significantly increased sustainable employability ($b = 0.11$, $p = 0.0297$). Furthermore, the unstandardized indirect effect of story condition on sustainable employability through awareness of PFWS was 0.05 (95% CI = .004 to .12). Therefore, H3 was supported.

H4 predicted that positive stories lead to higher sustainable employability than negative stories only when the participants identify with the main character. The moderation of identification with the main character on the impact of story on sustainable employability was significant ($b = 0.42$, $p < 0.001$). Fig 2 shows the interaction of identification with the main character and story type on sustainable employability. The simple slope of story condition for participants low in identification (-1 *SD*) was -.47, $p = 0.002$, showing that the negative story was related to higher sustainable employability than the positive story when identification with the main character was low. Moreover, in line with Hypothesis 4, the simple slope of story condition for participants high in identification (+ 1 SD) was .38, p = .021, showing that the positive story was related to higher sustainable employability than the negative story when identification with the main character was high. A graph with the impact of the interaction of identification with the main character and story type on sustainable employability. H4 is therefore supported.

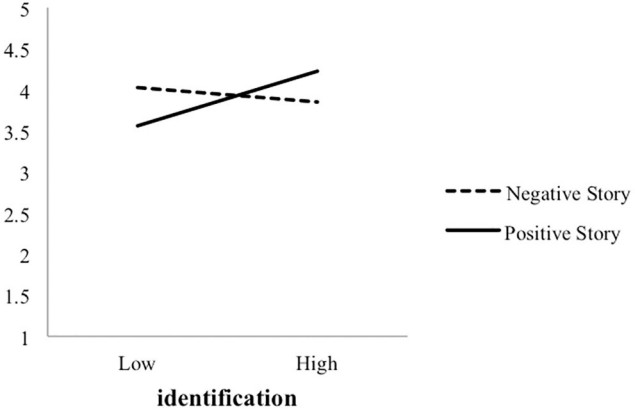

Moderation of identification with the main character on the relationship between story and sustainable employability. Low = 1 SD below the mean, high = 1 SD above the mean. For this graph the other parameters in the model were scored on the mean.

**Fig 2. Interaction between type of story and identification with main character in relation to sustainable employability.**

## Discussion

This study investigated whether reading a positive illness narrative about working with a chronic illness related to more sustainable employability of patients with the same condition, and whether this process was mediated by positive emotions and awareness of PFWS, and moderated by identification with the main character. A field experiment indicated that positive illness narratives only contributed to higher sustainable employability than negative illness narratives when people personally engaged with the narratives. First, positive stories significantly contributed to higher sustainable employability when they activated patients to think about their own desired future work selves. Second, the extent to which the person identified with the main character of the narrative determined whether the positive or the negative narrative was more related to high sustainable employability.

Based on socio-emotional selectivity theory [49] we expected that positive stories would increase people's perception of an open-ended future, and increase planning and a future-oriented mind-set, which are strongly linked to sustainable employability [14]. This would also be in accordance with the shifting perspectives model of chronic illness [50], whereby a positive story could make people switch from an illness perspective to a wellness perspective, whereby he or she focuses on other important life goals, including working life. Nonetheless, no direct relationship was found between reading a positive illness narrative and sustainable employability, indicating that merely reading a positive story does not automatically increase sustainable employability–more is needed to elicit this change.

In addition, contrary to our expectations, positive emotions did not mediate the impact between positive narratives and sustainable employability. We found that positive stories led to more positive emotions than negative stories, which is in line with emotional contagion mechanisms [31]. However, the proposed broad-and-build effects of these positive emotions [72] leading to an increased motivation and proactive mind-set towards one's own employability in the future did not occur. These results imply that merely experiencing a positive emotional state does not translate to concrete increased sustainable employability.

Awareness of desired PFWS did significantly mediate the relationship between reading a positive story and sustainable employability. This is in line with the idea that narratives offer people opportunities to experiment with different possible selves [53, 54] and with studies showing that positive and hopeful stories increase wellbeing and positivity [73]. Patients in the positive story condition had the opportunity to put themselves in the shoes of the main character, who successfully managed to work with IBD, and those who as a result became more aware of their own desired work selves, showed higher sustainable employability.

Furthermore, this study shows that the impact of positive illness narratives on fellow patients is moderated by the identification with the story character. In line with the Identification-Contrast model [33, 58], participants who strongly identified with the main character in the story seemed to assimilate the other's position, leading to higher sustainable employability when reading the positive rather than in the negative illness narrative. On the other hand, participants who did not identify with the character seemed to contrast themselves with the character in the story, leading to higher sustainable employability after reading a negative rather than a positive narrative (cf. [59]. This suggests that confrontation with a positive story that is regarded as unattainable for oneself, may sometimes lead to discouragement if identification is low. The moderation effect of identification is in line with research showing that in order for outstanding characters to be effective role models, identification with them is important [74].

"The significant relationships of both desired PFWS and identification with sustainable employability suggest that personal engagement with an illness narrative is necessary for its impact. With personal engagement we mean that the narrative has to resonate with the patient on a personal level in order to exert an influence, because the results indicate that merely reading the positive story–even when this induces positive emotions–does not relate positively to sustainable employability. The significant mediation of desired PFWS suggests that to imagine oneself in a desired work future is a prerequisite for perceiving a *sustainable* work future for oneself–a fundamental part of sustainable employability [14]. In addition, personal engagement can be linked to the notion of identification [75], whereby it takes effort or psychological involvement for the reader to put themselves in the shoes of a character. Identification and personal engagement could be further stimulated by increasing perceived similarity with a character, for example by having similar attitudes towards life, age or educational level [75]."

Furthermore, since positive emotions induced by a positive story did not contribute to higher sustainable employability, it is likely that cognitive processes are also required for an illness narrative to impact sustainable employability. Indeed, awareness of possible selves has been conceptualized as an interplay of emotional and (meta)cognitive mental processes [76, 77]. Since a possible self is fundamental to a person's current self-concept [78], awareness of a desired PFWS can have a strong motivating power [76–78] as was portrayed in the study of Strauss and colleagues [32]. Similarly, the interplay of emotional with cognitive mental processes related to awareness of possible selves, can explain the higher sustainable employability among patients with increased awareness of desired PFWS in comparison to those who only experienced a positive mood as a result of reading the positive story.

## Limitations and implications for future research

A number of limitations regarding the present study should be addressed. First, this experimental study aimed to explore the impact of illness narratives on patients with a chronic illness by investigating the immediate impact of one positive versus negative story on participants' sustainable employability. The duration of the effects of the story is therefore uncertain. Even though the results of this study suggest that a deeper, cognitive form of mental processing may be involved, future studies should investigate this in more depth and explore the impact of

exposure to several stories over time. Gaining more insight into in-depth and long-term mechanisms of narrative impact could also contribute a deeper understanding of at which moments of a patient's journey reading a positive narrative could exert the most beneficial impact, for example at the time of diagnosis, during a flare-up or at time of remission. Moreover, with the widespread availability of illness narratives online and in the media, future studies could also focus on how to counter the influence of negative narratives with positive experiences of fellow patients. This could also be applied in future studies, whereby patients in a negative story condition could besides receiving a positive mood manipulator also read the positive narratives in the end of the study. In addition, this study focused only on written narratives, future studies could explore other ways to present narratives (e.g. visual), or longer stories, such as books or novels and investigate the impact of actively discussing narratives in counseling sessions. Future research could also focus on intervention programs that combine narratives with other forms of occupational health therapy. In such interventions, personal engagement with positive role models could also be encouraged.

It also has to be noted, that although this is an experimental study, most of the findings are the result of internal analyses, relating measured rather than manipulated constructs. This precludes definite causal interpretation of our findings. For example, we cannot say for sure that sustainable employability really is the effect of identification with the positive story character, as people with a higher score for sustainable employability might identify more with the positive character to begin with. We tried to counter this by controlling for mental health and mastery, which we measured before reading the story. Future studies should, however, try to manipulate identification with the story character rather than merely measuring it.

Thirdly, this study focused on sustainable employability, which is a complex construct with several conceptualizations and different measurements [14, 79]. Nevertheless, research on sustainable employability has focused on people's subjective experience of their chances for obtaining or retaining long-term employment [14, 16]. In line with this research, we focused on the participant's perceptions of a common core element found in each of these conceptualizations: the ability and motivation to maintain healthy working life until retirement age (see also [14, 16, 51]. However, it remains unclear to what extent these conceptualizations of sustainable employability are related to actual prolonged successful employment. We believe this field of study would greatly benefit from long-term prospective studies that examine this relationship.

Additionally, by focusing on the perceptions of individuals, this study took a personal agency perspective [80] investigating whether patients can be empowered through positive illness narratives. Even though it was beyond the scope of this study, the work opportunity for patients provided by employers is also an important factor for achieving sustainable employment [14, 16]. In turn, inequality of the work environment has a profound impact on people's health [81] and for chronic patients this is even more important. Overall, patients with a chronic illness have lower employment ratings compared to the rest of the population [82]. Since intervention programs that stimulate employment for people with a chronic illness have shown to be effective [83], future research could expand the scope of this study by investigating the impact of positive work stories of chronic patients on employers' motivation to create opportunity for these employees.

Fifthly, this study focused on sustainable employability for chronic patients based on the rationale that employment provides people with both manifest (salary) and latent benefits (e.g. time structure, collective purpose and social contacts), which are linked to higher psychological well-being [9]. Even though this study mainly focused on paid employment, it should be noted that volunteering work or informal care for others, can also provide such latent benefits

[9]. Future studies could adopt a broader notion of work by including these types of participation more explicitly in their research.

Finally, this study specifically focused on the effect of illness narratives on patients with IBD, a group of chronic illnesses that has an increased incidence rate with more diagnoses every year [84]. Even though it is evident that every chronic illness will present its own challenges for sustainable employability, we aimed to uncover more general mechanisms by which positive illness narratives can inspire fellow patients. More research is needed to investigate to what extent results can be generalized to patients with other chronic illnesses.

### Practical implications

On the basis of this study, it could be advised to occupational health professionals and career counselors to use inspiring illness narratives for patients in order to increase their motivation and proactivity for a sustainable working life. This study indicates that it is important that patients can personally relate to the story. For example, a young student with IBD who wishes to study a semester abroad will be most empowered by hearing a story of another student with IBD who managed to study abroad despite of illness. In addition, more personal engagement and cognitive involvement with stories can be stimulated by for example more in-depth interaction with patients or by giving them exercises to actively think about desired PFWS.

### Conclusion

Improving the sustainable employability of individuals with a chronic illness is highly important for society, for employers and for the ill individuals concerned. Our study shows that reading narratives about successful fellow patients may contribute to sustainable employability. Moreover, our study shows that it is essential that such a narrative is personally relevant for the individual and promotes identification with the main character in the story. A positive narrative may then increase motivation and perceived ability to work until retirement as the narrative enhances a positive future work self.

### Supporting information

**S1 File. Illness narratives stimuli.**
(PDF)

**S2 File. PFWS question text.**
(PDF)

**S3 File. Sustainable employability scale.**
(PDF)

### Acknowledgments

The participants for this study were recruited through the online newsletter of the Dutch IBD foundation (CCUVN), with special thanks to Daniëlle van der Horst and Petra Tap.

### Author Contributions

**Conceptualization:** Inge M. Brokerhof, Jan Fekke Ybema, P. Matthijs Bal.

**Data curation:** Inge M. Brokerhof.

**Formal analysis:** Inge M. Brokerhof, Jan Fekke Ybema.

**Investigation:** Inge M. Brokerhof, Jan Fekke Ybema.

**Methodology:** Inge M. Brokerhof, Jan Fekke Ybema, P. Matthijs Bal.

**Project administration:** Inge M. Brokerhof, Jan Fekke Ybema.

**Supervision:** Jan Fekke Ybema, P. Matthijs Bal.

**Validation:** P. Matthijs Bal.

**Visualization:** Inge M. Brokerhof.

**Writing – original draft:** Inge M. Brokerhof.

**Writing – review & editing:** Jan Fekke Ybema, P. Matthijs Bal.

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
