## [Decision Letter · Decision Letter 0]

22 Oct 2019

PONE-D-19-26970

Illness narratives and chronic patients’ sustainable employability:  The impact of positive work stories

PLOS ONE

Dear Prof Brokerhof,

Thank you for submitting your manuscript to PLOS ONE. After careful consideration, we feel that it has merit but does not fully meet PLOS ONE’s publication criteria as it currently stands. Therefore, we invite you to submit a revised version of the manuscript that addresses the points raised during the review process.

Please see comments below. 

We would appreciate receiving your revised manuscript by 21 December 2019. To enhance the reproducibility of your results, we recommend that if applicable you deposit your laboratory protocols in protocols.io, where a protocol can be assigned its own identifier (DOI) such that it can be cited independently in the future. For instructions see: http://journals.plos.org/plosone/s/submission-guidelines#loc-laboratory-protocols

We look forward to receiving your revised manuscript.

Kind regards,

Andrew Soundy

Academic Editor

PLOS ONE

Journal Requirements:

1.

2.

We note that you have indicated that data from this study are available upon request. PLOS only allows data to be available upon request if there are legal or ethical restrictions on sharing data publicly. For more information on unacceptable data access restrictions, please see http://journals.plos.org/plosone/s/data-availability#loc-unacceptable-data-access-restrictions.

Additional Editor Comments (if provided):

Dear Author,

Please consider the comments made by reviewers and resubmit.

Reviewers' comments:

Reviewer's Responses to Questions

**Comments to the Author**

1. Is the manuscript technically sound, and do the data support the conclusions?

Reviewer #1: Yes

Reviewer #2: Yes

2. Has the statistical analysis been performed appropriately and rigorously? 

Reviewer #1: Yes

Reviewer #2: Yes

3. Have the authors made all data underlying the findings in their manuscript fully available?

Reviewer #1: Yes

Reviewer #2: Yes

4. Is the manuscript presented in an intelligible fashion and written in standard English?

Reviewer #1: Yes

Reviewer #2: Yes

5. Review Comments to the Author

Reviewer #1: The authors present a clear, logical argument backed by robust statistical analysis. The study is a useful addition to the literature. There are a few areas that require some clarification, but other than these this is a publishable manuscript. These clarification questions are listed by line number:

line 240 - please identify the control variables by name

line 325 - 82.5% were (not was)

line 349 - will the authors explain why bootstrapping was used/needed in their analysis?

line 393 and after - the authors use the term "personally engaged" and "personal engagement" several times - please define and clarify exactly what you mean by engagement - is this the same as the measurement "identification" and how was personal engagement measured?

line 276-277 - The authors are encouraged to include the text they used to describe for participants the PFWS and the prompt/question used to get participants to complete the micro-narrative.

Reviewer #2: Overall feedback:

This article makes a clear contribution to the study of chronic disease and illness narratives. In particular, the manuscript documents the important role of narrative and personal identification in people’s Possible Future Work Selves (PFWS). As such, the results of this study have the potential to positively impact medical patients who might be struggling to maintain employment throughout the lifespan.

Areas of strength:

• The introduction features an exemplar from a participant who read the positive story and experienced a positive outcome. This example dutifully set the stage for the rest of the study and is a helpful addition to the beginning of the manuscript.

• The manuscript has a clear flow of information and builds a strong argument.

• The study utilizes helpful theories that help frame and guide the work.

• The manuscript features a cogent review of literature.

• The study features four well-established hypotheses.

• The data collection and analysis sections are clear. I appreciate the strategic use of control variables.

Potential areas for improvement:

• Line 92: You might consider alternate wording of “shows influences of” to add some clarity. An example that might work is “is undergirded by”.

• Lines 133-136: You might clarify and expand on this sentence. For example, how was this influence perceived as inducing decision-making bias or engaging in affective forecasting?

• Line 248: It was wise to include a positive mood manipulator for participants who read the negative story. However, the story was very negative and discouraging. Therefore, you might reflect on additional ways you can potentially counteract the possible effects of a negative story. For example, would it be appropriate for participants to receive a positive story after completion of the study? You might consider adding this to the Limitations or Implications for Future Research if space allows.

• Lines 270-273: I’m not fully understanding these two sentences. Perhaps it would help to insert a sentence about negative emotions after the sentence of positive emotions. As is, the sentence about positive emotions is followed by a sentence on both positive and negative (in which emotions were loaded on separate constructs), so it felt like a bit of a jump.

• Discussion section: You might consider discussing how medical providers can logistically incorporate the positive narrative that the participants in this study utilized. For example, do you recommend that physicians share a printed story with a patient after initial diagnosis and/or at a key point in the patient’s journey (e.g., a time when they are struggling and/or have a flare up)? Do you recommend that the provider follow up with the patient during the next visit to see if they read the story and/or would like another printed copy of it? (or maybe an electronic copy if they would like to save and/or share it with others?)

Minor errors:

• Line 62: Insert a space in between “lives” and “(5)”.

• Line 69: Insert a space in between “well-being” and “(9)”.

• Line 114: “patient’s” should be “patients’ “

• Line 134: “patient’s” should be “patients’ “

• Line 186: Insert a space in between “(14)” and “We”

• Line 279: Insert a comma after “(32)”

• Line 287: Italicize “M” and “SD” to match the formatting that follows

• Line 293: All other sections referencing a Likert-scale read “(Answer with a 5-point Likert scale…)” so consider adding “Answer with a” to this sentence for consistency.

• Line 308: I think the word “exists” might be mistaken for “consists”

6. PLOS authors have the option to publish the peer review history of their article (what does this mean?). If published, this will include your full peer review and any attached files.

Reviewer #1: No

Reviewer #2: No

---

## [Author Response · Author response to Decision Letter 0]

23 Dec 2019

Reviewer Comments and our responses:

Reviewer #1: The authors present a clear, logical argument backed by robust statistical analysis. The study is a useful addition to the literature. There are a few areas that require some clarification, but other than these this is a publishable manuscript. These clarification questions are listed by line number:

line 240 - please identify the control variables by name

Thank you for this suggestion. We agree this needs clarification and have specified it in the manuscript. 

line 325 - 82.5% were (not was)

Thank you for noticing. We have changed the wording here. 

line 349 - will the authors explain why bootstrapping was used/needed in their analysis? 

Thank you for this suggestion, we will clarify this. Bootstrapping is advocated by Hayes (2012) to generate bias-corrected confidence intervals, correct for irregular shaped sampling distributions and increase the robustness of the indirect effect. We have used the standard bootstrap of 5,000 resamples in PROCESS (Hayes, 2012, p.12). 

In the method section, line 323 of the manuscript, we have added the following: “In order to generate bias-corrected confidence intervals, correct for irregular shaped sampling distributions and increase the robustness of the indirect effect, the standard bootstrap option of 5,000 resamples in PROCESS was used (71).”

We have also added the paper of Hayes (2012) to the reference list, which gives a concise explanation of the value bootstrapping in these types of analyses. 

line 393 and after - the authors use the term "personally engaged" and "personal engagement" several times - please define and clarify exactly what you mean by engagement - is this the same as the measurement "identification" and how was personal engagement measured? 

Thank you for this point! We have changed a discussion paragraph (lines 440 - 452 to explain this more clearly. Personal engagement is something that came out of the results: merely reading the story did not increase sustainable employability, some form of personal engagement (either via PFWS or identification) was necessary for the story to exert an influence. This is the new paragraph: 

“The significant relationships of both desired PFWS and identification with sustainable employability suggest that personal engagement with an illness narrative is necessary for its impact. With personal engagement we mean that the narrative has to resonate with the patient on a personal level in order to exert an influence, because the results indicate that merely reading the positive story – even when this induces positive emotions – does not relate positively to sustainable employability. The significant mediation of desired PFWS suggests that to imagine oneself in a desired work future is a prerequisite for perceiving a sustainable work future for oneself – a fundamental part of sustainable employability (14). In addition, personal engagement can be linked to the notion of identification (75), whereby it takes effort or psychological involvement for the reader to put themselves in the shoes of a character. Identification and personal engagement could be further stimulated by increasing perceived similarity with a character, for example by having similar attitudes towards life, age or educational level (75).”

line 276-277 - The authors are encouraged to include the text they used to describe for participants the PFWS and the prompt/question used to get participants to complete the micro-narrative.

Thank you for this suggestion. We have added the question text of Possible Future Work Self micro-narrative question in the appendix, in Supplement 2. 

Reviewer #2: Overall feedback:

This article makes a clear contribution to the study of chronic disease and illness narratives. In particular, the manuscript documents the important role of narrative and personal identification in people’s Possible Future Work Selves (PFWS). As such, the results of this study have the potential to positively impact medical patients who might be struggling to maintain employment throughout the lifespan.

Areas of strength:

• The introduction features an exemplar from a participant who read the positive story and experienced a positive outcome. This example dutifully set the stage for the rest of the study and is a helpful addition to the beginning of the manuscript.

• The manuscript has a clear flow of information and builds a strong argument.

• The study utilizes helpful theories that help frame and guide the work.

• The manuscript features a cogent review of literature.

• The study features four well-established hypotheses.

• The data collection and analysis sections are clear. I appreciate the strategic use of control variables.

Thank you for this feedback! 

Potential areas for improvement:

• Line 92: You might consider alternate wording of “shows influences of” to add some clarity. An example that might work is “is undergirded by”.

Thank you for this suggestion. We have changed the phrasing in “is undergirded by” as this better captures what we mean. 

• Lines 133-136: You might clarify and expand on this sentence. For example, how was this influence perceived as inducing decision-making bias or engaging in affective forecasting?

Many thanks for this suggestion. We have changed the paragraph (lines 133 – 140) to clarify this and elaborate a bit more o the decision-making bias. We have also added a source, which gives an overview of narrative communication in cancer prevention. Different studies have implied different results and narrative mechanisms. We hope to portray this more clearly in the revised text: 

“A variety of narrative impact mechanisms in medical contexts has been found, ranging from narratives increasing attention for and comprehension of medical information (45), narratives inducing decision-making bias by making people rely on heuristic rather than systematic mental processes (44), or, contradictorily, narratives have shown to stimulate patients to reflect on their own impact bias, thereby helping them overcome barriers for cancer screening, for example by making them aware of their own affective forecasting (40).”

• Line 248: It was wise to include a positive mood manipulator for participants who read the negative story. However, the story was very negative and discouraging. Therefore, you might reflect on additional ways you can potentially counteract the possible effects of a negative story. For example, would it be appropriate for participants to receive a positive story after completion of the study? You might consider adding this to the Limitations or Implications for Future Research if space allows.

We think the idea of countering the influence of a negative narrative with a positive narrative is valuable to add to the manuscript. We have added lines 474 – 478 in the section on Limitations and implications for future research:

“Moreover, with the widespread availability of illness narratives online and in the media, future studies could also focus on how to counter the influence of negative narratives with positive experiences of fellow patients. This could also be applied in future studies, whereby patients in a negative story condition could besides receiving a positive mood manipulator also read the positive narratives in the end of the study.”

• Lines 270-273: I’m not fully understanding these two sentences. Perhaps it would help to insert a sentence about negative emotions after the sentence of positive emotions. As is, the sentence about positive emotions is followed by a sentence on both positive and negative (in which emotions were loaded on separate constructs), so it felt like a bit of a jump

Thank you for this suggestion, we have changed the wording of this paragraph (lines 274 – 280) to make this clearer. This is the revised text:

“Participant’s emotions after reading the story were measured by a list of 12 positive and 12 negative emotions, whereby people were asked to select all emotions that they currently experienced, based on the Multiple Affect Adjective Checklist (MAACL, (62, 63). Examples of items referring to positive emotions were “encouraged”, “inspired” or “comforted”, and examples of items referring to negative emotions were “sad”, “discouraged”, “irritated” or “anxious”. The positive emotions captured two constructs, namely: comfort and inspiration, whereas the negative emotions captured three constructs, namely: depression, hostility and anxiety. In the present study only positive emotions were examined, and comport and inspiration were combined into one score for positive affect.”

• Discussion section: You might consider discussing how medical providers can logistically incorporate the positive narrative that the participants in this study utilized. For example, do you recommend that physicians share a printed story with a patient after initial diagnosis and/or at a key point in the patient’s journey (e.g., a time when they are struggling and/or have a flare up)? Do you recommend that the provider follow up with the patient during the next visit to see if they read the story and/or would like another printed copy of it? (or maybe an electronic copy if they would like to save and/or share it with others?)

Thank you for this suggestion! We feel it is indeed important to add this point in the section on Limitations and implications for future research as this could be worthwhile to investigate. We have changed lines 467 – 473 to include this in revised manuscript:

“The duration of the effects of the story is therefore uncertain. Even though the results of this study suggest that a deeper, cognitive form of mental processing may be involved, future studies should investigate this in more depth and explore the impact of exposure to several stories over time. Gaining more insight into in-depth and long-term mechanisms of narrative impact could also investigate at which moments of a patient’s journey reading a positive narrative could exert the most beneficial impact, for example at the time of diagnosis, during a flare up or at time of remission.”

Minor errors:

• Line 62: Insert a space in between “lives” and “(5)”.

• Line 69: Insert a space in between “well-being” and “(9)”.

• Line 114: “patient’s” should be “patients’ “

• Line 134: “patient’s” should be “patients’ “

• Line 186: Insert a space in between “(14)” and “We”

• Line 279: Insert a comma after “(32)”

• Line 287: Italicize “M” and “SD” to match the formatting that follows

• Line 293: All other sections referencing a Likert-scale read “(Answer with a 5-point Likert scale…)” so consider adding “Answer with a” to this sentence for consistency.

• Line 308: I think the word “exists” might be mistaken for “consists”

Thank you so much for your detailed feedback and suggestions. We have changed all the errors you marked. We have changed “Answer with a 5-point Likert scale” consistently in “Responses on a 5-point Likert scale”. 

Added References based on the Reviews:

Hayes, A. F. (2012). PROCESS: A versatile computational tool for observed variable mediation, moderation, and conditional process modeling.

Kreuter MW, Green MC, Cappella JN, Slater MD, Wise ME, Storey D, et al. Narrative communication in cancer prevention and control: a framework to guide research and application. Annals of behavioral medicine. 2007;33(3):221-35.

---

## [Decision Letter · Decision Letter 1]

21 Jan 2020

Illness narratives and chronic patients’ sustainable employability:  The impact of positive work stories

PONE-D-19-26970R1

Dear Dr. Brokerhof,

We are pleased to inform you that your manuscript has been judged scientifically suitable for publication and will be formally accepted for publication once it complies with all outstanding technical requirements.

With kind regards,

Andrew Soundy

Academic Editor

PLOS ONE

Additional Editor Comments (optional):

Reviewers' comments:

Reviewer's Responses to Questions

**Comments to the Author**

1. If the authors have adequately addressed your comments raised in a previous round of review and you feel that this manuscript is now acceptable for publication, you may indicate that here to bypass the “Comments to the Author” section, enter your conflict of interest statement in the “Confidential to Editor” section, and submit your "Accept" recommendation.

Reviewer #1: All comments have been addressed

Reviewer #2: All comments have been addressed

2. Is the manuscript technically sound, and do the data support the conclusions?

Reviewer #1: (No Response)

Reviewer #2: (No Response)

3. Has the statistical analysis been performed appropriately and rigorously? 

Reviewer #1: (No Response)

Reviewer #2: (No Response)

4. Have the authors made all data underlying the findings in their manuscript fully available?

Reviewer #1: (No Response)

Reviewer #2: (No Response)

5. Is the manuscript presented in an intelligible fashion and written in standard English?

Reviewer #1: (No Response)

Reviewer #2: (No Response)

6. Review Comments to the Author

Reviewer #1: (No Response)

Reviewer #2: Thank you for the many dutiful updates you have made to the manuscript. This was already a strong piece of work and is now ready for publication. I look forward to seeing this in print.

7. PLOS authors have the option to publish the peer review history of their article (what does this mean?). If published, this will include your full peer review and any attached files.

Reviewer #1: No

Reviewer #2: No

---

## [Editor Report · Acceptance letter]

24 Jan 2020

PONE-D-19-26970R1 

Illness narratives and chronic patients’ sustainable employability: The impact of positive work stories 

Dear Dr. Brokerhof:

I am pleased to inform you that your manuscript has been deemed suitable for publication in PLOS ONE. Congratulations! Your manuscript is now with our production department. 

With kind regards,

on behalf of

Dr. Andrew Soundy 

Academic Editor

PLOS ONE